# Changes in Gait Parameters and the Podal System Depending on the Presence of a Specific Malocclusion Type in School-Age Children

**DOI:** 10.3390/jcm12237334

**Published:** 2023-11-26

**Authors:** Dorota Różańska-Perlińska, Jarosław Jaszczur-Nowicki, Łukasz Rydzik, Jacek Perliński, Joanna M. Bukowska

**Affiliations:** 1Medical Department, The Academy of Applied Medical and Social Sciences, 82-300 Elbląg, Poland; dorotka@gumed.edu.pl (D.R.-P.); jacekperlinski@wp.pl (J.P.); 2Department Physiotherapy, School of Public Health, Collegium Medicum, University of Warmia and Mazury, 10-719 Olsztyn, Poland; joanna.bukowska@uwm.edu.pl; 3Institute of Sports Sciences, University of Physical Education, 31-571 Krakow, Poland

**Keywords:** malocclusion, gait parameters, foot pressure, stomatognathic system

## Abstract

Background: The correspondence between various aspects of human postural parameters and the spatial relation of the jaws is of increasing interest among scientists. Emerging research suggests that the stomatognathic system and posture play, in a broad sense, significant roles. Objectives: The aim of the study was to analyze the relationship between various malocclusion types and gait parameters, the distribution of foot pressure on the ground, and body balance. Methods: The study involved 155 patients aged 12-16. The subjects were divided into groups according to their malocclusion—Angle’s class II (*n* = 32), Canine class II (*n* = 31), and Overbite (*n* = 46). The control group (*n* = 46) comprised children not demonstrating any defects. The study data were collected by direct observation of the oral cavity. Gait analysis was carried out using the Wiva^®^ Science sensor, and the distribution of foot forces on the ground and body balance was determined via the E.P.S R/1 pedobarographic mat. The non-parametric Mann–Whitney U test was applied for statistical analysis. Results: Analysis of the results obtained showed statistically significant differences in left step duration (*p* = 0.042) and the duration of the right step (*p* = 0.021), as well as the projection of the body’s center of gravity on the left foot (*p* = 0.027). Conclusions: Distoocclusion in the anterior part of the mandible may cause different positioning of the head and neck, as well as varying tension of the muscles, further leading to balance disorders while walking.

## 1. Introduction

In recent years, researchers have been increasingly interested in understanding the connections between various aspects of human postural parameters and the spatial relation of the jaws. Formerly traditionally considered as independent fields of study, nowadays, the relationship between the podal system and malocclusion is gaining the attention of many investigators [1,2,3]. In emerging studies, it has been suggested that the stomatognathic system and posture play, in a broad sense, significant roles. Michalakis found that occlusal instability is associated with changes in lateral body mass distribution [4], while Perez-Belloso determined a relationship between the presence of Angle’s class II and a forward movement of the center of gravity [5].

The physiological correspondence among occlusion and the two fundamental movements performed in everyday life—standing and walking—seems to be a complex set of many contributing factors. Julia-Sanchez indicated different biomechanical and viscoelastic properties of selected muscles in relation to different malocclusal traits [6]. Furthermore, Iacob recorded differences in plantar pressure on the external sensors of the ipsilateral foot and the internal sensors of the contralateral foot with increased values while experimentally inducing malocclusion in patients [7]. Researchers not only point out changes in podal balance in association with the relation of the jaws, but also reveal numerous variations in the gait cycle, including its stability and rhythm. Fujimoto investigated the effect of different mandibular positions on gait rhythm by examining 12 young, healthy individuals walking for 18 m in six different positions with occlusal splints. It was concluded that a change in mandibular position could affect gait stability [8].

Understanding the statistically important connection between gait parameters and malocclusion traits has significant implications for both orthodontists—providing insight into the underlying etiologies of occlusal disorders and leading to a search for new, potentially more effective treatment methods—and physiotherapists, who may contribute to the identification of patients at risk of developing malocclusion and the introduction of early prevention. For example, Mason indicates the effect of RPE (rapid palatal expansion) on posture in children with regard to gait, in which it becomes improved after correction of the arches’ transverse dimensions in the cranio-caudal direction [9].

On the other hand, it should be emphasized that malocclusion is a multifactorial condition influenced by environmental and genetic elements. Also, the gait cycle can be determined by various factors, including neuromuscular control, the musculoskeletal and cardiorespiratory systems, as well as age, personality, and sociocultural factors.

Our research team conducted the first stage of an investigation regarding correspondence between the podal system, gait parameters, and malocclusion. In the examination, gait, the distribution of foot pressure on the ground, and body balance were examined. The study group consisted of 76 patients aged 12–16 years and was divided into two categories—without malocclusion and with malocclusion. Angle’s classification was used, which enabled the determination of the anteroposterior relationship of the first molars. A pedobarographic mat was used to analyze the distribution of foot forces on the ground. The Wiva^®^ Science diagnostic system was used for gait analysis and Kineod 3D was implemented for posture analysis. The performed research allowed us to prove that there is a relevant connection between the presence of stomatognathic disorders and walking rhythm, gait cycle duration, and right step duration time. Patients with malocclusion showed a high-speed walking rhythm and a decreased duration of the gait cycle, as well as of the right step. Furthermore, students with a proper relation between the jaws (Angle’s class I) presented a low-speed rhythm and an increased duration of the gait cycle and of the right step [10]. Analysis of the obtained results and the conclusions that we came to led us to investigate the issue once again, taking different types of anteroposterior malocclusion into consideration using Angle’s classes I, II, and III; Canine classes I, II, and III; and also vertical plane disorders including deep and open bites. Moreover, we decided to examine a larger population of children in order to make the results potentially more relevant. The field of interest in the research can provide important indications that will allow orthodontic practitioners to modify and improve existing algorithms for the treatment of malocclusions, e.g., enriching them with physiotherapy or specially selected physical exercises. The aim of the study was to analyze the relationship between various types of malocclusion and gait parameters, the distribution of foot pressure on the ground, and body balance. The hypothesis statement assumes that there is a relationship between dental abnormalities in malocclusion and some gait parameters, foot pressure on the ground and body balance, especially in the dominant part of the body (right foot).

## 2. Materials and Methods

The research data were collected from 155 patients. In total, 50.3% of them (78 students) were female, and 49.7% (77 students) were male. All the students were between the age of 12 and 16 (x = 13.07 SD = 1.15). The study group was acquired from regular patients of the Twoje Zdrowie EL Health Centre who came for a dental check-up and who agreed, by informed written consent from their parents or guardians, to be included in the investigation. Parents/guardians were previously given information about the research, and they filled out a questionnaire with the necessary data as required. The study group consisted of students from four different schools; two of them were located in the city, while the other two were in the countryside.

The following exclusion criteria were adopted: previous lower limb or upper body surgery, previous severe trauma altering the child’s initial posture, previous orthodontic and/or orthotic treatment, insufficient teeth to determine dental classification of occlusion and/or skeletal disorders.

The investigation was accepted by the Ethics Committee for Scientific Research of the University of Warmia and Mazury, Olsztyn (Decision No. 9/2018).

Patients were examined using Angle’s and Canine classifications which allow for the determination of the anterioposterior relation of the jaws, and accordingly, the relation of the first upper and lower molars and of the upper and lower canine teeth. The anterioposterior relationship of the jaws (Angle’s and Canine classifications) can be arranged into three classes: I—normal occlusion, II—distoocclusion, and III—mesioocclusion.

### 2.1. Angle’s Classes Can Be Described

Class I—The mesiobuccal cusp of the maxillary first molar aligns with the mesiobuccal groove of the first mandibular molar. The mesial incline of the maxillary canine occludes with the distal incline of the mandibular canine. The distal incline of the maxillary canine occludes with the mesial incline of the first mandibular premolar;Class II—The mesiobuccal groove of the first mandibular molar is DISTALLY (posteriorly) positioned when in occlusion with the mesiobuccal cusp of the first maxillary molar. The mesial incline of the maxillary canine occludes ANTERIORLY with the distal incline of the mandibular canine. The distal surface of the mandibular canine is POSTERIOR to the mesial surface of the maxillary canine by at least the width of a premolar;Class III—The mesiobuccal cusp of the maxillary first permanent molar is occluded DISTALLY (posteriorly) to the mesiobuccal groove of the first mandibular molar. The distal surface of the mandibular canines is mesial to the mesial surface of the maxillary canines by at least the width of a premolar. The mandibular incisors are in complete crossbite. Moreover, the following malocclusion features were observed: cross bites, lingual occlusion, as well as deep and open bites. The study data were collected by direct observation of the oral cavity [11].

There were 46 patients with Angle’s class I and the participants with this type comprised the control group. There were 32 students diagnosed with Angle’s class II and 11 children with Angle’s class III.

### 2.2. Canine Classification Can Be Characterized

Class I: Mesial incline of the upper canine overlaps with the distal slope of the lower canine (the maxillary canine occludes between the mandibular canine and the first premolar);Class II: Distal slope of the maxillary canine occludes or contacts the mesial slope of the lower canine;Class III: The mandibular canine is displaced anteriorly to the maxillary canine with no overlapping [12].

There were 46 children in the control group—Canine class I. Among the rest, 31 were diagnosed with Canine class II and 5 with Canine class III.

In mixed dentition worldwide, the distributions of malocclusion classes I, II, and III were 73% [40–96%], 23% [2–58%], and 4% [0.7–13%] [13]. Due to this fact, groups of patients with Angle’s class III (11 children) and Canine class III (5 children) were excluded from the statistical comparison due to the risk of lowering the credibility of the results.

Furthermore, vertical plane discrepancies were taken into consideration. Deep bite malocclusion is described as the “overlapping of the upper incisors on the labial surface of the lower incisors vertically, when the standard limit of 1–2 mm is exceeded” [14]. Anterior open bite is the lack of contact between the incisal edges of the maxillary and mandibular anterior teeth in centric relation [15]. The research group included 46 patients with an overbite and three with an open bite. Children diagnosed with a lack of contact between the incisors were excluded from the statistical comparison because the sample was too small to be reliable. The study data were collected by direct observation of the oral cavity.

### 2.3. Instruments

Body mass was measured using the Tanita InnerScan^®^V model BC-545N (Tanita Corporation, Maenocho, Itabashiku, Tokyo, Japan). Body height was estimated via the Soehnle electronic stadiometer (Soehnle, Gaildorfer Straße 6, 71522 Backnang, Germany).

The Wiva^®^ Science diagnostic sensor (Letsens Group, Letsens S.R.L. Via Buozzi, CastelMaggiore, Bologna, Italy) was used to analyze gait, and the E.P.S R/1 pedobarographic mat (Letsens Group, Letsens S.R.L. Via Buozzi, CastelMaggiore) was applied to measure foot pressure on the ground, Bologna, Italy). Both of the devices automatically transferred the collected data to the BioMech Studio software (Biomech Studio 2.0 Manual, Letsens Group, Letsens SRL Via Buozzi, CastelMaggiore, Bologna, Italy).

### 2.4. Procedure

In order to acquire parental consent for the experiment, the study schedule was presented and a specially developed form was used. The experimental sessions were carried out in a building that housed a dental practice, which was naturally lit. Participants were given privacy to avoid outside influences. Each person was tested individually and all tasks were performed under identical conditions during one session (ambient temperature: 22 °C). For data collection, each participant was anonymously registered in the BioMech Studio 2.0 program with the following data: participant code, date of birth, gender, body mass and height, which was measured by the researcher using a height-measuring device while maintaining an upright posture. The test procedure is shown in Figure 1.

### 2.5. Measurement Protocol

First, the bite of each participant was established by a dentist. Then, body height was measured using the Soehnle electronic stadiometer. Body mass was estimated using the Tanita InnerScan^®^V model BC-545N device. In both cases, the participant stood still with his/her arms along the trunk. The next stage comprised measurements on the E.P.S R/1 pedobarographic mat. The participants had to stand still on it for 20 s, with his/her hands along the trunk and eyes directed straight ahead. The study was conducted barefoot. The researcher then placed the Wiva^®^ Science sensor on the participant at L5 height so that the sensor did not move while walking. The task of the participants was to walk a 20 m section twice with natural gait.

### 2.6. Statistical Analysis

First, the Shapiro–Wilk test was used to analyze the distribution, which showed non-compliance with the normal distribution for all measurement parameters. Therefore, the non-parametric Mann–Whitney U test was used in further analyses. The level of significance in the study was set at *p* < 0.05. The Statistica program (StatSoft Polska, Kraków, Poland, version 13.3) was used to perform the statistical analysis.

## 3. Results

Statistical analysis of the collected data on gait parameters showed that there were statistically significant differences between the Canine class II (KKII) group and the control (GK). A comparison of the duration of the left step (Figure 2) and that of the right step (Figure 3) between the group with KKII malocclusion and the GK group is presented. Comparing both parameters, which are related to each other, it can be seen that a statistically significant longer left step time and shorter right step time—compared to the control group—were characteristic of the Canine class II group. The obtained results also indicated statistically significant differences at the level of *p* = 0.042 during the left step and *p* = 0.021 during the right step. Other parameters in the comparison of both groups and comparisons between Angle’s class II and GK, as well as the Overbite and GK groups, did not show statistically significant differences.

In Table 1, an assessment of the significance is shown concerning parameters of distribution in relation to the plantar pressure of the foot between the group with specific malocclusion and the control group. The comparison of the group with defects in the arrangement of the upper and lower canines and the control group did not show any statistically significant differences in any of the examined parameters. There were no statistically significant differences in the deep bite group compared to the control group, neither were there any noted between the Angle’s class and the control group.

The analysis of body balance between Angle’s class and GK, as well as the Canine class and the GK groups, did not show statistically significant differences in any of the examined parameters. The comparison of the Overbite group with the control showed statistically significant differences at the level of *p* = 0.027 in the projection of the center of gravity of the body on the left foot. There were no statistically significant differences in the analyzed parameter of the body’s center of gravity or in the right foot (Table 2).

## 4. Discussion

The relation between gait and the position of the jaws has been an intriguing issue for many researchers. It has been proved in more and more studies that there is an existing impact of malocclusion on posture parameters. For example, Silvestrini-Baviati underlines that postural, orthoptic, and occlusal alternations may be clinically associated [16]. Another scientist—Sambataro, proved that there is a correlation between scoliosis and malocclusions on the transversal plane [17]. In a systemic review by Stancker, the author analyzes 13 articles and confirms the thesis that there is an evident correspondence between human body posture and dental occlusion [18].

Undoubtedly, posture disorder may affect gait, but thus far, only one report has been found concerning the analysis of correlations between malocclusion and gait-parameter disturbances. Nowak, in 2023, found no effect of malocclusion distinguished with molar teeth position on the biomechanical characteristics of gait. Despite differences in the examined populations in the studies (Nowak examined 90 adult patients [19], whilst our investigation was carried out on 155 children and growing adolescents), in both investigations, it has been proved that disorders of Angle classification have no influence on gait parameters. This fact may suggest that there is no correlation between body balance and orthodontic disorders found in the posterior segment of the jaws. Abnormality in the position of the posterior teeth while maintaining the proper position of anterior teeth did not cause any disorders of gait parameters in this study.

However, in the literature, no studies have been found to date in which a correlation would be determined between gait characteristics and individual anterior–posterior and vertical types of malocclusion.

The gait examination of the patients diagnosed with Canine class II demonstrated that there is a statistically relevant difference in the duration of the right step and left step in comparison to students without orthodontic disorders. The duration of the left step was increased, while the duration of the right step was decreased in Canine class II. Such changes may suggest that distoocclusion in the front section of the jaw may cause different positioning of the head and neck, and moreover, their different muscle tension leading to balance disorders during walking. This impairing body coordination may potentially result in modification of the body’s gravity center and consequently, in the gait pattern. Canine class II patients make prolonged left steps. Then, they take shortened right steps. It may be assumed that the prolonged left step is taken with the weaker foot in order to compensate for the shortened right step taken with the dominant foot. On the other hand, this may also suggest that distalization of the front section of the lower jaw—Canine class II—causes a disturbance of the center of gravity, which results in firstly taking the shortened right foot step and subsequently, a longer left foot step in order to recompense the initial shorter step. On the contrary, when starting with the left foot, a longer left step in Canine class II is taken compared to Canine class I and subsequently, body balance is interrupted. As a result, the next step—right foot step, must be shortened in order to avoid losing balance. The interpretation of the above considerations may be twofold because in the study, the researchers did not define the participants’ dominant foot. Yet, the assumption was made that the difference between the right and left step duration is the result of the right foot being dominant. Still, further research is needed to confirm this hypothesis and gain more reliable results.

The subsequent group of students diagnosed with a deep bite showed the following variation in statistical analysis: increased barycenter of the left foot. These disturbed parameters confirm the hypothesis that there is a relationship between the anterior segment of the jaws and gait characteristics. There were no significant deviations in step duration and gait rhythm. The interesting fact is that there was no correlation between the deep bite and barycenter of the right foot (assumedly, the dominant foot), but there was a remarkable correspondence between overbite and left foot parameters (assumedly, the weaker foot). This thought seems to be connected with Michalakis’s observation on the influence of dental occlusion on body mass distribution changes. The research team concluded that clenching and occlusal stability are associated with changes in lateral body mass distribution [4]. Therefore, overbite, which is in fact a condition of maximally clenching the teeth in the vertical plane, could also induce the modification of mass distribution observed in the increased barycenter of the left foot (assumedly, the weaker foot).

Considering the increased barycenter of the left foot in patients with a deep bite, it could be concluded that students with this orthodontic condition have improved their stability during walking regarding the left foot. The research carried out by another author—Amaricai, allows us to confirm that the static plantar pressure is influenced by the position of the jaws in the maximum intercuspidation and opening of the mouth. They proved that maximum intercuspidation with relaxed masticatory muscles causes increased loading on the left foot and improved postural stability in comparison to maximum mouth opening [20]. The results of the study by Amaricai may suggest that, similarly, a deep bite with maximum intercuspidation causes improved stability, while an open bite could be the reason of increased loading on left foot. Yet, in order to verify the hypothesis, further research is needed.

Among 46 students identified with a deep bite, 14 of them were also diagnosed with Angle’s class II. For one of them, Angle’s class III was noted, while 16 of them were diagnosed with Canine class II, and 3 of them with Canine class III. Thus, the fact that some types of malocclusion may coexist in one patient must be taken into consideration. In the study, 13 children demonstrated an isolated overbite with Angle’s and Canine class I, which is 28.3% of the research group. Due to the complexity of orthodontic disorders, it is difficult to interpret particular gait parameters in relation to compounded malocclusion. Further research is needed to investigate the issue. It would probably be useful to distinguish isolated deep bite to acquire more explainable results.

It should not be forgotten that occlusion, as well as posture and podal parameters, are influenced by numerous factors. Research by Feka provided evidence that the weight status of the children can affect the plantar load distribution, with the obese category being different from other categories [21]. Moreover, Jaszczur-Nowicki [22] and Bukowska [23,24] proved that prolonged overloading with backpacks affects movement patterns, which may further lead to the acquisition and consolidation of postural defects. Therefore, the subject of this investigation requires more criteria to be taken into account in order to obtain more reliable and precise results.

## 5. Conclusions

There was a significant relationship between dental abnormalities in malocclusion and step duration. When analyzing children with Canine class II, a significant relationship was noticed between the duration of the left and right step.There were no statistically significant relationships between the pressure on the forefoot, midfoot, or heel area, as well as malocclusion (Angle’s class, Canine class, Overbite).There was a statistically significant relationship between the projection of the body’s center of gravity on the left foot and dental disorders in the group of patients with a deep bite.

## Figures and Tables

**Figure 1 jcm-12-07334-f001:**
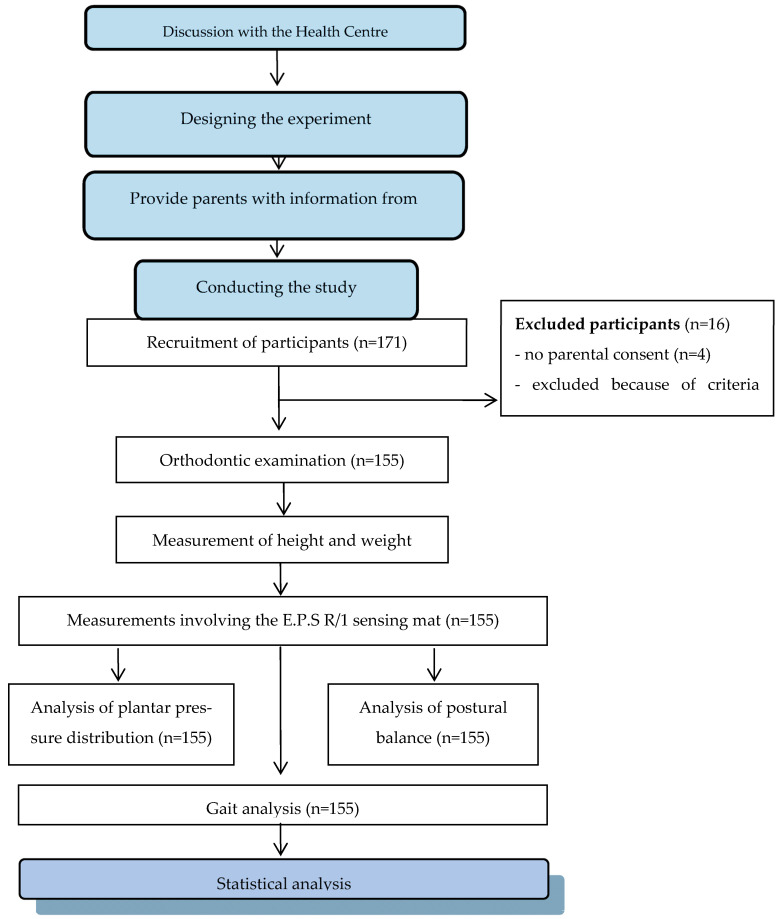
Scheme of the test procedure.

**Figure 2 jcm-12-07334-f002:**
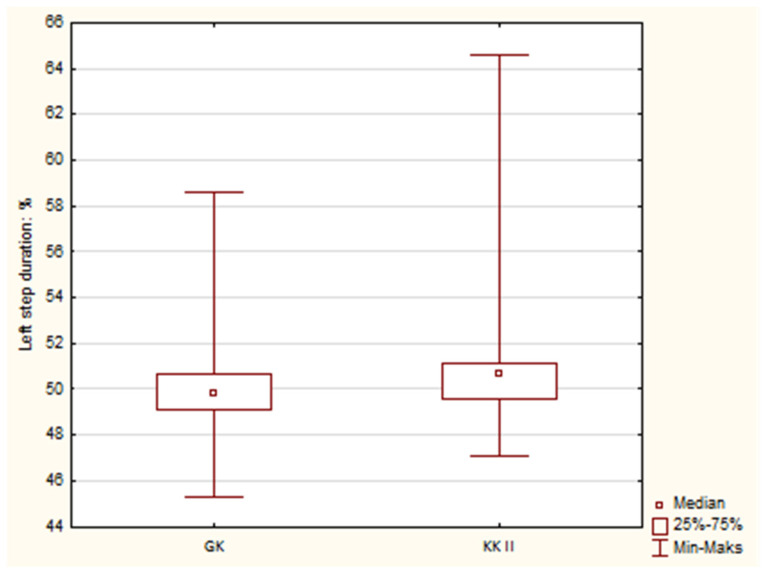
Box mustache chart comparing the left step duration in the Canine class II group with the control. GK—Control group, KKII—Canine class II.

**Figure 3 jcm-12-07334-f003:**
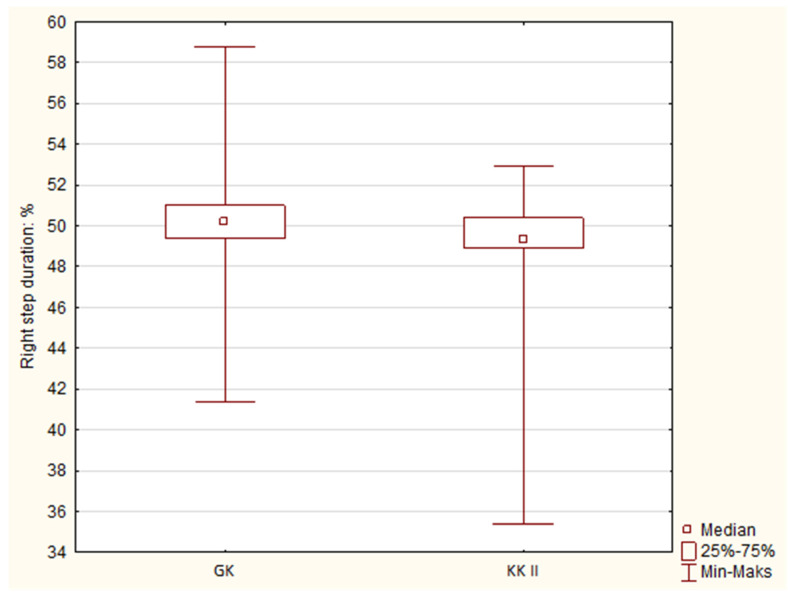
Box mustache chart comparing the duration of the right step in the Canine class II group with the control. GK—Control group, KKII—Canine class II.

**Table 1 jcm-12-07334-t001:** Significance of differences in the parameters of foot pressure distribution between the group with a specific defect and the control.

	Angle’s Class		Canine Class		Overbite	
	Z	*p*	Z	*p*	Z	*p*
Left foot load	1.000597	0.317022	1.308742	0.190623	0.417778	0.67611
Right foot load	−1.41709	0.156458	−1.7346	0.082812	−0.41387	0.678967
Forefoot LF [%]	−0.7568	0.449172	−0.75824	0.448308	−0.50368	0.61449
Metatarsus LF [%]	−1.28503	0.198782	−1.17371	0.240511	−1.02687	0.304481
Heel LF [%]	1.132656	0.25736	0.862108	0.388629	0.913644	0.360904
Forefoot RF [%]	−0.19301	0.846952	−0.44663	0.655139	−0.34359	0.731153
Metatarsus RF [%]	0.685689	0.49291	1.15294	0.248936	0.702803	0.482179
Heel RF [%]	−0.47236	0.636668	−0.54012	0.589118	−0.54272	0.587323

LF—left foot, RF—right foot, Z—U Mann–Whitney test. *p*—significance level. *p* ≤ 0.05.

**Table 2 jcm-12-07334-t002:** Significance of differences in balance parameters between the group with a specific defect and the control group.

	Angle’s Class		Canine Class		Overbite	
	Z	*p*	Z	*p*	Z	*p*
Body COP [mm^2^]	−1.23424	0.217115	−1.05426	0.291763	−1.72187	0.085094
COP LF [mm^2^]	−1.68121	0.092724	−1.87483	0.060818	−2.21383	0.026841
COP RF [mm^2^]	−0.61458	0.538832	−0.74266	0.457689	−0.88631	0.375449

COP—center of pressure, LF—left foot, RF—right foot, Z—U Mann–Whitney test. *p*—significance level. *p* ≤ 0.05.

## Data Availability

The data used in this study will be made available upon reasonable request by the author.

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
