# Peer review of "Changes in Gait Parameters and the Podal System Depending on the Presence of a Specific Malocclusion Type in School-Age Children"

_jcm, 2023, doi:10.3390/jcm12237334_

Round 1
Reviewer 1 Report
Comments and Suggestions for Authors
Dear authors,
The manuscript is very interesting and i have just a few comments that i believe will improve the paper. This research show a quantitative investigation of the relation between the presence of a specific type of malocclusion in school-age children and the changes in gait parameters and podal system. I consider the topic relevant, but i think that the authors should be some changes before, to be clear some lack of the paper.
The introduction section should present a greater number of references, since the limit of references is not a problem. Some of the studies that are recommended are the following:
- Cuccia A, Caradonna C. The relationship between the stomatognathic system and body posture. Clinics (São Paulo). (2009) 64:61–6. 10.1590/S1807-59322009000100011
-Cabrera-Domínguez ME, Domínguez-Reyes A, Pabón-Carrasco M, Pérez-Belloso AJ, Coheña-Jiménez M, Galán-González AF. Dental Malocclusion and Its Relation to the Podal System. Front Pediatr. 2021 Jun 22;9:654229. doi: 10.3389/fped.2021.654229.
-Michelotti A, Buonocore G, Manzo P, Pellegrino G, Farella M. Dental occlusion and posture: an overview. Prog Orthod. (2011) 12:53–8. 10.1016/j.pio.2010.09.010
The Materials and Methods section is adequate, but the figure 1 must be redone according to prisma standards. The CONSORT flow diagram should be referenced in the paper.
The result section show a text erro, line 202. Tables and figures must be presented according to the journal's standards.
Conclusion must responde to the main aim of the study. It is advisable that conclusion be clear and concise. Therefore, it is not recommended that it is lenght be so long. 3-5 line. The references must be reviewed completely and especially revised the numbers: 11, 17, 19.
Author Response
Dear Reviewer,
Thank you very much for your time and valuable comments, which all have been considered and incorporated. The detailed list of responses is given below. We hope that the modifications and explanation will be acceptable for you.
Yours sincerely,
Rydzik, corresponding author
The manuscript is very interesting and i have just a few comments that i believe will improve the paper. This research show a quantitative investigation of the relation between the presence of a specific type of malocclusion in school-age children and the changes in gait parameters and podal system. I consider the topic relevant, but i think that the authors should be some changes before, to be clear some lack of the paper.
A: Thank you
The introduction section should present a greater number of references, since the limit of references is not a problem. Some of the studies that are recommended are the following:
- Cuccia A, Caradonna C. The relationship between the stomatognathic system and body posture. Clinics (São Paulo). (2009) 64:61–6. 10.1590/S1807-59322009000100011
-Cabrera-Domínguez ME, Domínguez-Reyes A, Pabón-Carrasco M, Pérez-Belloso AJ, Coheña-Jiménez M, Galán-González AF. Dental Malocclusion and Its Relation to the Podal System. Front Pediatr. 2021 Jun 22;9:654229. doi: 10.3389/fped.2021.654229.
-Michelotti A, Buonocore G, Manzo P, Pellegrino G, Farella M. Dental occlusion and posture: an overview. Prog Orthod. (2011) 12:53–8. 10.1016/j.pio.2010.09.010
A: This has been corrected
The Materials and Methods section is adequate, but the figure 1 must be redone according to prisma standards. The CONSORT flow diagram should be referenced in the paper.
A: This has been corrected
The result section show a text erro, line 202. Tables and figures must be presented according to the journal's standards.
A: This has been corrected
Conclusion must responde to the main aim of the study. It is advisable that conclusion be clear and concise. Therefore, it is not recommended that it is lenght be so long. 3-5 line. The references must be reviewed completely and especially revised the numbers: 11, 17, 19.
A: This has been corrected
Reviewer 2 Report
Comments and Suggestions for Authors
This study investigates different types of malocclusions and their potential influence on balance disorders during walking among school-age children. Some revisions should be done as follows:
Introduction:
Line 50 – 51, line 54, it seems to be some additional space in between the text, please update accordingly throughout the whole manuscript.
The author mentioned the previous research findings in the related field, while the current gap and the rationale for conducting this research need to be strengthened.
At the end of the introduction, the author only mentions the aim of the study, the hypothesis should also be added to this section.
Results:
Line 202, the text in this line should be deleted.
Table 2, the author only provides the mean value for each variable, the standard deviation will also be needed here.
Discussion:
Lines 264 – 268, the author mentions the distalization of the front lower jaw causes disturbance of the center of gravity. More detail connections and the mechanisms between them need to be discussed and that would be the main concern of the study.
Author Response
Dear Reviewer,
Thank you very much for your time and valuable comments, which all have been considered and incorporated. The detailed list of responses is given below. We hope that the modifications and explanation will be acceptable for you.
Yours sincerely,
Rydzik, corresponding author
Introduction:
Line 50 – 51, line 54, it seems to be some additional space in between the text, please update accordingly throughout the whole manuscript.
A: This has been corrected
The author mentioned the previous research findings in the related field, while the current gap and the rationale for conducting this research need to be strengthened.
A: This has been corrected
At the end of the introduction, the author only mentions the aim of the study, the hypothesis should also be added to this section.
A: This has been corrected
Results:
Line 202, the text in this line should be deleted.
A: This has been corrected
Table 2, the author only provides the mean value for each variable, the standard deviation will also be needed here.
A: This has been corrected
Discussion:
Lines 264 – 268, the author mentions the distalization of the front lower jaw causes disturbance of the center of gravity. More detail connections and the mechanisms between them need to be discussed and that would be the main concern of the study.
A: This has been corrected
Round 2
Reviewer 1 Report
Comments and Suggestions for Authors
Once it has been verified that the authors have made the changes and improvements to the document. I think it is appropriate to publish it. congratulations
Reviewer 2 Report
Comments and Suggestions for Authors
The author has addressed all my questions and comments.